# A Confirmatory and an Exploratory Factor Analysis of the Cohen-Mansfield Agitation Inventory (CMAI) in a European Case Series of Patients with Dementia: Results from the RECage Study

**DOI:** 10.3390/brainsci13071025

**Published:** 2023-07-03

**Authors:** Bruno Mario Cesana, Eleni Poptsi, Magda Tsolaki, Sverre Bergh, Alfonso Ciccone, Emmanuel Cognat, Andrea Fabbo, Sara Fascendini, Giovanni B. Frisoni, Lutz Frölich, Maria Cristina Jori, Patrizia Mecocci, Paola Merlo, Oliver Peters, Carlo Alberto Defanti

**Affiliations:** 1Department of Clinical Sciences and Community Health, Unit of Medical Statistics, Biometry and Bioinformatics “Giulio A. Maccacaro” Faculty of Medicine and Surgery, University of Milan, 20122 Milan, Italy; brnmrcesana@gmail.com; 2Laboratory of Psychology, Section of Cognitive and Experimental Psychology, Faculty of Philosophy, School of Psychology, Aristotle University of Thessaloniki (AUTh), 54124 Thessaloniki, Greece; 3Greek Association of Alzheimer’s Disease and Related Disorders (GAADRD), 54643 Thessaloniki, Greece; tsolakim1@gmail.com; 41st Department of Neurology, School of Medicine, Aristotle University of Thessaloniki (AUTh), 54124 Thessaloniki, Greece; 5Research Centre for Age-Related Functional Decline and Disease, Innlandet Hospital Trust, 2313 Ottestad, Norway; sverre.bergh@sykehuset-innlandet.no; 6Norwegian National Centre for Aging and Health, Sykehuset i Vestfold, 3103 Tønsberg, Norway; 7Department of Neurology with Neurosurgical Activity “Carlo Poma” Hospital, ASST di Mantova, 46100 Mantua, Italy; alfonso.ciccone@asst-mantova.it; 8Cognitive Neurology Centre, Lariboisière-Fernand Widal Hospital GHU AP-HP Nord, 75010 Paris, France; emmanuel.cognat@aphp.fr; 9Geriatric Service-Cognitive Disorders and Dementia, Department of Primary Care, Local Health Authority of Modena (AUSL), 41124 Modena, Italy; a.fabbo@ausl.mo.it; 10FERB Alzheimer Centre, 24025 Gazzaniga, Italy; sara.fascendini@gmail.com (S.F.); carloalberto.defanti@ferbonlus.com (C.A.D.); 11Division of Geriatrics and Rehabilitation, University Hospitals of Geneva, 1205 Geneva, Switzerland; giovanni.frisoni@gmail.com; 12Department of Geriatric Psychiatry, Central Institute of Mental Health, Medical Faculty Mannheim, Heidelberg University, 68159 Mannheim, Germany; lutz.froelich@zi-mannheim.de; 13Mediolanum Cardio Research, 20123 Milan, Italy; jori@mcr-med.com; 14Institute of Gerontology and Geriatrics, Department of Medicine and Surgery, University of Perugia, 06123 Perugia, Italy; patrizia.mecocci@unipg.it; 15Division of Clinical Geriatrics, NVS Department, Karolinska Institutet Stockholm, 17177 Stockholm, Sweden; 16Neurological Unit (PM), U.V.A. Centre, Humanitas Gavazzeni, 24125 Bergamo, Italy; paola.merlo@gavazzeni.it; 17Department of Psychiatry, Charité-Universitätsmedizin Berlin, Campus Benjamin Franklin, 12203 Berlin, Germany; oliver.peters@charite.de

**Keywords:** RECage study, CMAI, confirmatory factor analysis, exploratory factor analysis, new model of scoring

## Abstract

Background: One of the most widely used instruments for assessing agitation in dementia patients is the Cohen-Mansfield Agitation Inventory (CMAI), nevertheless no global score has been proposed. The aim of this study is: (a) to conduct a confirmatory (CFA) and exploratory factor analysis (EFA) of CMAI on people with dementia and Psychological and Behavioral Symptoms (BPSD), and (b) to propose an alternative structure, based on clinical criteria including all CMAI items. Methods: Confirmatory and exploratory factor analyses were carried out on the CMAI 29 items administered at baseline to 505 patients with dementia (PwD) and BPSD enrolled in the international observational RECage study. Results: The three-factor structure has not been confirmed by the CFA, whilst the EFA was carried out respectively on 25 items disregarding 4 items with a prevalence ≤5% and then on 20 items disregarding 9 items with a prevalence ≤10%. The four-factor structure explaining 56% of the variance comprised Physically Aggressive behavior, Verbally Aggressive behavior, Physically non-aggressive behavior, and Physically and verbally aggressive behavior. Conclusions: A new grouping of all items according to a clinical criterion is proposed, allowing for a more sensible evaluation of the symptoms leading to better differentiation.

## 1. Introduction

Agitation is a manifestation of various behaviors that characterize many neuropsychiatric disorders and syndromes, including cognitive impairment and dementia. A commonly accepted description does not exist, but the International Psychogeriatric Association (IPA) has recently attempted to set the criteria of agitated behavior by following the structure of the DSM-5 criteria (2013) [1,2]. According to the consensus definition, agitation describes a situation where: (a) it occurs during a neurocognitive disorder, such as cognitive impairment or dementia; (b) for a minimum of two weeks, the patient expresses behaviors associated with emotional distress, classified in three general categories (excessive motor activity, verbal aggression, and physical aggression); (c) the behaviors are severe enough to cause significant impairment in interpersonal relationships, in social functioning, as well as in activities of daily living; and (d) besides the fact that the co-morbidity of agitation with other conditions could be present, agitation is not attributable solely to another psychiatric, medical, or psychological condition [2].

Agitation during dementia is a common behavioral symptom observed in such patients of any etiology, such as of dementia due to Alzheimer’s disease [3], due to Lewy Body [4] or due to frontotemporal dementia [5]. Agitation can be found in any stage of the course of dementia, from very mild to mild, and from moderate to severe, either in patients living at nursing homes [6,7] or in community-dwelling patients [8] and it is usually associated with increased healthcare costs [6]. According to a recent review, there is a wide prevalence of agitation symptoms in dementia reported in the literature, with prevalence rates between 5% to 88% that vary by geographic region, and with lower ranges to be reported for Asia [9]

Until today, several rating scales that are utilized in people with dementia to assess neuropsychiatric symptoms and agitation exist. Such commonly utilized scales in clinical settings are the Neuropsychiatric Inventory (NPI) [10], the Neurobehavioral Rating Scale (NBRS) [11], the Behavioral Pathology in Alzheimer’s Disease Rating Scale (BEHAVE-AD) [12], or the Pittsburgh Agitation Scale (PAS) [13]. 

One of the most widely used instruments for assessing agitation in patients with dementia is the Cohen-Mansfield Agitation Inventory (CMAI) designed by Cohen-Mansfield et al., in 1989 [14]. Initially, the instrument was developed for use at nursing homes, but during the past years it was also used in various clinical settings. Different versions exist, including the short or the long form of the instrument, the long form with expanded definitions, as well as the community form and the disruptiveness form [15]. 

The original (long form) of the CMAI rating questionnaire contains one question for each of the following 29 agitation behaviors: (1) pace, aimless wandering, (2) inappropriate dress or disrobing, (3) spitting, (4) cursing or verbal aggression, (5) constant unwarranted request for attention or help, (6) repetitive sentences or questions, (7) hitting (including self), (8) kicking, (9) grabbing onto people, (10) pushing, (11) throwing things, (12) strange noises (weird laughter or crying), (13) screaming, (14) biting, (15) scratching, (16) trying to get to a different place, (17) intentional falling, (18) complaining, (19) negativism, (20) eating/drinking inappropriate substances, (21) hurt self or other, (22) handling things inappropriately, (23) hiding things, (24) hoarding things, (25) tearing things or destroying property, (26) performing repetitious mannerisms, (27) making verbal sexual advances, (28) making physical sexual advances, and (29) general restlessness. Regarding scoring, there is a 7 points Likert rating scale for each behavior, ranging from 1 to 7 (1 = never, 7 = several times an hour), indicating the observation frequency of each behavior. The CMAI inventory requires approximately 20 min to be completed [15]. 

As far as the psychometric properties are concerned, many studies in several countries attempted assessing the validity of the CMAI using factor analysis. Given the great diversity of the behaviors CMAI contains, constructors decided not to calculate a total score (by adding all items), but to identify different factors. One of the first validation studies was performed in three units of a nursing home and included 66 residents (15 men and 51 women, age range from 59 to 96 years) [16]. This study identified three factors of agitation: aggressive behavior (including hitting, kicking, pushing, scratching, tearing things, cursing or verbal aggression, grabbing onto people, biting, and spitting); physically non-aggressive behavior (pacing, inappropriate robing or disrobing, trying to get to a different place, handling things inappropriately, general restlessness, and repetitious mannerisms), and verbally agitated behavior (complaining, repeated requests for attention, negativism, repetition of sentences or questions, and screaming) [14,16]. The scores of the three aforementioned factors were considered along with the absolute and relative frequencies of the agitated or not agitated status for each factor. It is important to remark that the factors did not include all items of the CMAI rating scale, since items with low prevalence were disregarded. Other studies have also indicated a three-factor model that explains the behavioral symptoms [14,17,18,19] but with different items considered. 

Similarly, the study of Choy et al. [17] enrolled inpatients and outpatients (164 people with dementia, 57 men and 107 women with an age range from 55 to 102 years) from two hospital units in Hong Kong (Castle Peak Hospital and the Prince of Wales Hospital). According to their factor analysis, three different factors were identified: (a) physically aggressive behaviors, (b) physically non-aggressive behaviors, and (c) verbally agitated behaviors, while six of the twenty-nine behaviors were excluded from the analysis since they rarely occurred, being present in less than 5% of the subjects. 

Some validation studies were performed in community-based populations. One of the first studies on a community-based sample was performed by Cohen-Mansfield [20]. According to that exploratory and confirmatory analysis, there was evidence for three factors (verbally agitated behavior, verbally-non agitated behavior, and physically non-aggressive behavior) in both staff and relatives’ ratings, whilst an extra factor of physically aggressive behavior was suggested in the relatives’ ratings. Due to insufficient data, the authors did not come to a conclusion as to whether a three- or a four-factor model was most appropriate for capturing the broad range of agitation. 

Subsequently, Koss et al., (1997) conducted research on an English-speaking community-based population of 306 persons, 241 of which had Alzheimer Disease. In this study, an expanded and revised list of 38 items of the CMAI version was utilized. The two additional items derived from directly observable behaviors (temporal occurrence of agitation) and one open item was added as well (possible additional disruptive behavior). The analysis identified four factors: (1) physically non-aggressive behaviors, (2) physically aggressive behaviors, (3) verbally non-aggressive behaviors, and (4) verbally aggressive behaviors. However, the aforementioned four factors do not include all the 38 items of the community CMAI rating scale, since some had a low prevalence [21]. 

Similarly, Rabinowitz, Davidson, De Deyn, Katz, Brodaty, and Cohen-Mansfield, in 2005, investigated a large sample of 1265 older people with BPSD from three nursing homes by using the 29 items nursing home version. This analysis also identified four factors: (a) Aggressive Behavior (hitting, kicking, scratching, biting, pushing, grabbing, throwing things, cursing or verbal aggression, spitting, tearing things/destroying property, hurting self or others, screaming), and (b) Physically Non-Aggressive Behavior (pacing, trying to get to a different place, general restlessness, inappropriate dressing or disrobing, handling things inappropriately, performing repetitious mannerisms), (c) Verbally Agitated Behavior (complaining, constant requests for attention, repetitive questions, negativism), and (d) Hiding and Hoarding. In his study, like in others, items with low frequency (less than 9%), were excluded from the analysis. These items were: eating inappropriate substances, physical sexual advances, verbal sexual advances, and intentional falling [22].

More recently, Patrick et al., investigated a cohort of 609 persons with dementia, followed up by a memory clinic and their caregivers [23]. The researchers excluded from the analyses the CMAI items occurring in less than 5% of caregivers. The excluded items were: (a) intentionally falling, (b) hurting the self or others, (c) spitting, (d) scratching, (e) pushing, (f) biting, (g) kicking, (h) making physical and verbal sexual advances, (i) and eating/drinking inappropriate substances. The results of the exploratory factor analysis initially supported a seven-factor structure that explained 62% of the variance. However, based on the studies suggesting that the number of items in each factor should be greater than or equal to three [22,24], the number of the extracted factors was limited to three. Thus, three robust factors accounted for 52.45% of the total variance, and were (a) physically aggressive, (b) physically non-aggressive, and (c) verbally agitated. 

There are also a few studies conducted with patients living in care facilities. Schnelli, Ott, Mayer, & Zeller (2021), studied the data retrieved from the nursing documents of 1182 clients seeking services of six home care organizations in Switzerland in patients with dementia, delirium, and other psychiatric disorders. In their study the factor analysis revealed five main factors, these being (a) searching behaviors, (b) physically aggressive behaviors, (c) disruptive behaviors, (d) verbally aggressive behaviors, and (e) importunate behaviors. Nevertheless, in this study the participants included not only people with dementia but also with other psychiatric disorders [25]. Finally, a relatively recent study highlighted the validity of an observation-based version of the CMAI (CMAI-O), which has proven to be a promising research tool for independently measuring agitation in people living with dementia in Nursing Home settings. This tool provides additional information based not only on proxy informants but also on direct observation that helps to better define the symptoms labeled as “agitation” and follow their changes over the time. This “new” tool could represent an interesting line of research and a new possibility in clinical practice [26].

### Objective of the Study

The aim of our study was to conduct a confirmatory and exploratory factor analysis of the long form of the CMAI inventory according to the manual [15] on people with dementia and Behavioral and Psychological Symptoms of Dementia (BPSD) enrolled in the RECage clinical trial [27]. 

We especially considered the problem of excluding items from the analysis due to their low prevalence. In fact, by excluding items with low frequency, apart from losing useful information, we would risk not being able to distinguish between patients with or without these behaviors, ultimately not reaching a valid assessment of the patient’s severity. 

The study was based on all CMAI factors stated in the CMAI manual [15], since according to it, it is not useful to calculate a total score by adding the individual scores of all items. Former researchers decided to exclude some items from the factor analyses based on their low prevalence (usually ≤5% or ≤10%.) and avoiding the consequent statistical problems occurring in the factor analysis calculation.

Therefore, the main aim of the study is to propose an alternative (more clinical) factor structure of the CMAI that includes all of its items.

## 2. Materials and Methods

### 2.1. Design and Procedure

The present study was part of the REspectful Caring for AGitated Elderly study (RECage). The RECage study was a longitudinal and multicultural study comprising 11 clinical centres from six European countries, (2018–2023). The main aim of the RECage project was to evaluate the short- and long-term clinical efficacy of the Special Medical Care Units (SCU-B) for people with dementia and Behavioral and Psychological Symptoms of Dementia (BPSD). A SCU-B was defined as a “residential medical structure lying outside of a nursing home, in a general hospital or elsewhere, e.g., in a private hospital or a geriatric or psychiatric hospital, where patients with BPSD are temporarily admitted when their behavioral disturbances are not amenable to control at home” [27]. Therefore, this study compared the first cohort of PwD followed by SCU-B with the possibility of being admitted for copying BPSD, and the second cohort (control-non SCU-B) without having this possibility. In comparison with the non SCU-B units, by means of the pattern of the NPI total score over the follow-up time of 36 months; the pattern of the three CMAI factors was moved from a co-primary objective to a secondary one with an ad hoc protocol amendment.

The RECage study also had secondary and tertiary objectives, such as assessing the Quality of Life of the patients and their caregivers, the cost-effectiveness of SCU-Bs, and psychotropic drug consumption. Finally, the assessment of SCU-Bs’ capacity to delay the time to Nursing Home Placement (NHP) was also an objective. For more details, please see Poptsi et al., 2021 [27]. 

### 2.2. Participants

The total sample of the RECage study amounted to 508 patients. Three patients were excluded because of CMAI missing data. Therefore, the final study sample consisted of 505 patients. The participants were followed up every 6 months for three years by 11 European clinical centers (6 non-SCU-B and 5 SCU-B) [27]. 

Of the enrolled patients, 279 were females (54.9%), with a mean age of 78.1 years (SD = ±7.95), and a mean education of 8.93 (±4.53) years. The mean score of the Mini Mental State Examination (MMSE) [28] was 15.4 (±6.25), whilst the mean score in Neuropsychiatric Inventory (NPI) [10] was 52.5 (±18.97). 

The CMAI factor means were 12.2 (±4.77) for Factor 1 “Aggressive Behavior”, 17.1 (±8.17) for Factor 2 “Physically Non-aggressive Behavior”, and, finally, 16.5 (±7.30) for Factor 3 “Verbally Agitated Behavior”. It must be noted that these mean scores characterize a low involvement (9–63 is the interval of the values for Factor 1 with 9 items, 6–42 for Factor 2 with 6 items and, finally, 5–35 for Factor 3 with 5 items). Furthermore, the mean NPI score was lower than half of its maximum total value of 144. Moreover, there was an intermediate involvement for the MMSE with values ranging from 0 to 30 and we have to take into account that the 38.0% of the patients were categorized as “moderately severe/severely impaired AD patients” (<15), and the 37.8% were categorized as “Moderately impaired AD patients” (15–20 extremes included). Regarding the main relevant part of the protocol, it is presented in Poptsi et al. [27]. 

### 2.3. Instruments

The battery of tests administered in the RECage study was: (1) for the assessment of the general cognitive status the MMSE [28], (2) for the assessment of the general functional status the Activities of Daily Living scale (ADL) [29], (3) for the assessment of neuropsychiatric symptoms the NPI [10], and (4) for the assessment of the patient’s BPSD the Cohen-Mansfield Agitation Inventory (CMAI) [15]. Additional scales for the assessment of the quality of life were also administered [27].

### 2.4. The CMAI

For the needs of the present study, the long CMAI questionnaire was used, which is the original version used in nursing home’s population, containing 29 items. 

The main reason for utilizing the long form of the 29-item CMAI instead of the 37-item Community form (CMAI-C) was to reduce the burden of the caregivers in the RECage study and to reduce the time of each visit attended by outpatients. Therefore, the three-factor structure of the CMAI nursing home shown in its manual has been taken into consideration.

### 2.5. Statistical Methods

Descriptive statistics have been calculated for quantitative variables (mean, standard deviation) and for qualitative (categorical) variables (absolute and percent frequency).

Confirmatory and exploratory factor analyses (CFA and EFA, respectively) have been carried out with PROC CALIS and PROC FACTOR of SAS^®^ Version 9.4, respectively. The CFA hypothesis is that a 3-factor structure is evident when considering the 20 items analyzed in the factor analysis reported in the CMAI manual. The adequacy of fitting has been assessed by the χ2 test and the values of the Root Mean Square Error of Approximation (RMSEA < 0.05 for a good model fit), the Standardized Root Mean Square Residual (SRMSR < 0.05 for a good model fit, and, finally, by the Bentler’s comparative fit index (>0.90 for a good model fit). EFA has been carried out on a different set of items, depending on their prevalence, aiming to obtain the most parsimonious factor structure with a clear clinical interpretation.

Several rotation methods (varimax, promax, etc.) have been used in EFA after the factor extraction with the Principal Component Analysis (PCA) to obtain a better differentiation of the factor loadings [30,31,32].

## 3. Results

Nine items were recorded in less than 10% of our case series. Five of them (“Hurt self or others”, “Making physical sexual advances”, “Intentional falling”, “Eating/drinking inappropriate substances”, and “Making verbal sexual advances”) were excluded from the factor analysis according to the CMAI manual [15]. Four items (“Kicking”, “Biting”, “Scratching”, and “Spitting”) in our case series showed a prevalence lower than 10%. The low prevalence of some items lead us to disregard them from the factor analysis, in order to find a clinically sensible structure of their rating scale (Table 1). 

### 3.1. Confirmatory Factor Analysis (CFA)

After 12 iterations the convergence has been reached. The model fit χ2 was 960.156 (df = 167, *p* ≤ 0.0001), statistically rejecting the confirmatory factor model of the CMAI. Indeed, the Root Mean Square Error of Approximation (RMSEA) was 0.2289, greater than the conventional 0.05 value for a good model fit. The Standardized Root Mean Square Residual (SRMSR) was 0.0867, not close to the conventional 0.05 value for a good model fit. In addition, Bentler’s comparative fit index was 0.7262, much lower than the required value of at least 0.90, leading us to conclude that the model was very poorly fitted. So, taking into account that all the above four criteria consistently testify to an inadequate model fit, it is possible to conclude that the CMAI three-factor model proposed in the CMAI manual was not adequate for our data.

### 3.2. Exploratory Factor Analysis (EFA)

(1) The EFA carried out on all 29 CMAI items did not reach a computational result owing to the error that the maximum number of iterations has been exceeded.

(2) Therefore, we were obliged to discard from the analysis the items “14—Biting” (2.57%), “15—Scratching” (3.96%), “17—Intentional falling” (NC) (4.55%), and “28—Making physical sexual advances (NC)” (5.94%) with a prevalence lower than 6%. It must be noted that the items “17—Intentional falling”, and “28—Making physical sexual advances” were also not considered in the three-factor structure of the CMAI reported in the CMAI manual [15]. Furthermore, our analysis included the following seven items “11—Throwing things (14.65%)”, “12—Strange noises (weird laughter or crying) (33.27%)”, “20—Eating/drinking inappropriate substances (9.70%)”, “21—Hurt self or other (6.14%)”, “23—Hiding things (50.20%)”, “24—Hoarding things (41.39%)”, and “27—Making verbal sexual advances (7.52%)”, not considered in the CMAI structure, with three factors reported in the CMAI manual [15]. The four-factor structure of the CMAI, is presented in Table 2.

The variance explained by each factor before the “varimax” rotation was 5.069 (56%) for Factor 1, 2.127 (23%) for Factor 2, 1.323 (15%) for Factor 3, and 0.737 (8%) for Factor 4. After the “varimax” rotation, the variance explained by each factor was 2.824 (31%) for Factor 1, 2.565 (28%) for Factor 2, 2.441 (27%) for Factor 3, and 1.234 (14%) for Factor 4.

Factor 1 includes “Physically Aggressive behaviors” and comprises the following 6 items: “7—Hitting (including self) (F1)”, “8—Kicking (F1)”, “9—Grabbing onto people (F1)”, “10—Pushing (F1)”, “11—Throwing things (NC)”, and “21—Hurt self or other (NC)”. These items practically correspond to CMAI Factor 1 “Aggressive behavior” with the addition of items 11 and 21 not considered in the Factor analysis reported in the CMAI manual.

Factor 2 presents “Verbally Aggressive behaviors” and it is constituted by the following 6 items: “5—Constant unwarranted request for attention or help (F3)”, “6—Repetitive sentences or questions (F3)”, “12—Strange noises (weird laughter or crying) (NC)”, “18—Complaining (F3)”, “19—Negativism (F3)”, and “29—General restlessness (F2)”. These items practically correspond to CMAI Factor 3 “Verbally agitated behavior” with the addition of item 12 not considered in the Factor analysis reported in the CMAI manual. It must be noted that maybe the item “29—General restlessness (F2)” with a relevant load on this factor could be more pertinent to this factor than to the CMAI Factor 2.

Factor 3 comprises the following 9 items, which mostly include “Physically non-aggressive behaviors”: “1—Pace, aimless wandering (F2)”, “2—Inappropriate dress or disrobing (F2)”, “16—Trying to get to a different place (F2)”, “20—Eating/drinking inappropriate substances (NC)”, “22—Handling things inappropriately (F2)”, “23—Hiding things (NC)”, “24—Hoarding things (NC)”, “25—Tearing things or destroying property (F1)”, and “26—Performing repetitious mannerisms (F2)”. These items practically correspond to CMAI Factor 2 “Physically non-aggressive behavior” with the addition of items 20, 23, and 24 not considered in the Factor analysis reported in the CMAI manual. However, item “25—Tearing things or destroying property (F1)” with a prevalence of 14.06% loads very similarly also on Factor 1 (0.26794 and 0.29928, respectively). This item belongs to CMAI Factor 1 “Aggressive behavior”.

Factor 4 is constituted by “Physically and verbally aggressive behaviors” and consists of the following 4 items: “3—Spitting (F1)”, “4—Cursing or verbal aggression (F1)”, “13—Screaming (F3)”, and “27—Making verbal sexual advances (NC)”.

In addition, a further Factor Analysis on the 20 items with a prevalence greater than 10% has been carried out. However, only two items “4—Cursing or verbal aggression (F1)” and “13—Screaming (F3)” load on Factor 4, and, in addition, only three items “10—Pushing (F1)”, “11—Throwing things (NC)”, and “25—Tearing things or destroying property (F1)” load on Factor 3. Moreover, the latter item loads very similarly on Factor 2 and Factor 3 (0.24364 and 0.25429, respectively).

Therefore, we repeated the above Factor Analysis on the 20 items with a prevalence greater than 10% by constraining the Factor number to three. Thus, the items “3—Spitting” (7.33%), “8—Kicking (8.91%), “20—Eating/drinking inappropriate substances (NC)” (9.70%), “21—Hurt self or others (NC) (6.14%)”, and “27—Making verbal sexual advances (NC) (7.52%)” have been excluded. It must be noted that items “20—Eating/drinking inappropriate substances (NC)”, “21—Hurt self or others (NC), and “27—Making verbal sexual advances (NC)” were not considered in the factor analysis with three factors described in the CMAI manual (Table 3).

The variance explained by each factor was 4.602 (62.7%) for Factor 1, 1.748 (23.8%) for Factor 2, and 0.992 (13.5%) for Factor3. After the “varimax” rotation, the variance explained by each factor was 2.602 (35.4%) for Factor 1, 2.421 (32.9%) for Factor 2, and 2.320 (31.6%) for Factor 3.

We define Factor 1 as “Verbally Aggressive behavior” which comprises episodes of verbal abuse, without causing any physically harmful events to other people. Factor 1, is constituted by the following 6 items: ”5—Constant unwarranted request for attention or help (F3)”, 6—Repetitive sentences or questions (F3)”, “12—Strange noises (weird laughter or crying) (NC)”, “18—Complaining (F3)”, “19—Negativism (F3)”, and ”29—General restlessness (F2)”. These items practically correspond to CMAI Factor 3 “Verbally agitated behavior” with the addition of item 12 not considered in the Factor analysis reported in the CMAI manual. It has to be noted that maybe the item “29—General restlessness (F2)” with a relevant load on this factor could be more pertinent to this factor than to the CMAI Factor 2.

Then, we define Factor 2 as “Physically Aggressive behavior”, since it consists of severe aggressive behaviors, including physical force intending to harm another person or damage an object. According to other research, aggression in persons with dementia does not have an intent, but is rather an expression of anxiety, fear, cognitive decline and confusion. In any case, physically aggressive behavior is considered to lead to hospitalization of people with BPSD, as well as to physical and psychological distress of the caregivers. Therefore, the “Physically Aggressive behavior”, according to our results, is constituted by the following 7 items: “4—Cursing or verbal aggression (F1)”, “7—Hitting (including self) (F1)”, “9—Grabbing onto people (F1)”, “10—Pushing (F1)”, 11—Throwing things (NC)”, ”13—Screaming (F3)”, and 25—Tearing things or destroying property (F1). These items practically correspond to CMAI Factor 1 “Aggressive behavior” with the addition of item 11 not considered in the Factor analysis reported in the CMAI manual and of item “13—Screaming (F3)” considered in the CMAI Factor 3 “Verbally agitated behavior”.

However, item “25—Tearing things or destroying property (F1)” with a prevalence of 14.06% also loads very similarly on Factor 3 (0.26152 and 0.25112, respectively).

Finally, we name Factor 3 as “Physically non-aggressive behavior” since it consists of annoying behaviors expressed physically, which are not physically harmful either for the patient or others. According to our results the “Physically non-aggressive behavior” was constituted by the following 7 items: “1—Pace, aimless wandering (F2)”, “2—Inappropriate dress or disrobing (F2)”, “16—Trying to get to a different place (F2)”, “22—Handling things inappropriately (F2)”, “23—Hiding things (NC)”, “24—Hoarding things (NC)”, and “26—Performing repetitious mannerisms (F2)”. These items belong to CMAI Factor 2 “Physically non-aggressive behavior” with the addition of the items 23 and 24 not considered in the Factor analysis reported in the CMAI manual.

## 4. Discussion

We did not successfully confirm the three-factor structure of the 29-item CMAI. At the same time, we confirmed that it is not possible to include all 29 items of the CMAI in the factor analysis, since we obtained a computational error of having exceeded the maximum number of iterations from the output of the PROC FACTOR of SAS due to the lower prevalence (≤5%, at least) of some items. Furthermore, the exploratory analysis of 25 items with a prevalence greater than 6% led to a four-factor CMAI structure, which comprised 6 items loading to factor one, 6 items loading to factor two, 9 items loading to factor three, and 4 items loading to factor four. 

Our results are in partial agreement with the CMAI exploratory analysis results, described in the CMAI manual and referred to community samples. To be accurate, it seems that 4 of the 6 items that loaded to our Factor 1 correspond to the “Aggressive behavior” (according to the CMAI manual), five of the six items loading on our Factor 2 correspond to CMAI Factor 3 “Verbally agitated behavior”, and, finally, six items of the nine items loading on our Factor 3 practically correspond to CMAI Factor 2 “Physically non-aggressive behavior” (according to the CMAI manual). In contrast to the baseline factor analyses of CMAI in nursing homes, in our study an extra factor (Factor 4) was revealed which included four items; three items (spitting, cursing or verbal aggression, and screaming) of these, four items are included in the CMAI Factor 1 (“Aggressive behavior”) whilst the item “Making verbal sexual advances” was not considered in the initial CMAI factor analysis in nursing homes shown in the CMAI manual. However, the fact that these three items are already considered in the CMAI Factor 1 “aggressive behavior” limits the relevance of our finding, allowing us to conclude that a further subdivision into four factors cannot have any clinical relevance. 

Our study is also in agreement with other studies which suggest the presence of four factors of CMAI [20,21,33]. 

Moreover, we attempted to conduct a further factor analysis on the twenty items with a prevalence greater than 10%. Therefore, nine items (spitting kicking, biting, scratching, intentional falling, eating/drinking inappropriate substances, hurt self or other, making verbal sexual advances, and making physical sexual advances) were excluded. It must be remarked that three of the items we excluded (eating/drinking inappropriate substances, hurt self or others, and making verbal sexual advances) were also excluded from the factor analysis with three factors shown in the CMAI manual. 

The three constrained factors found were: (1) Factor 1 comprised six items, of which four corresponded to Factor 3 of CMAI, which is “Verbally agitated behavior”, with the exception of the items of strange noises (weird laughter or crying) that are not considered in the analysis according to the CMAI manual, as well as the item of general restlessness that could be more pertinent to this factor than to the CMAI Factor 2. (2) Out of the seven items of Factor 2, five corresponded to CMAI Factor 1 which was “Aggressive behavior”, whilst the item throwing things was not considered by the Factor analysis reported in the CMAI manual, while the item “screaming” was considered in the CMAI Factor 3 “Verbally agitated behavior”. It is worth mentioning that the item “tearing things or destroying property” which loads to Factor 1 of the CMAI manual was also loading on Factor 3. (3) Finally, Factor 3 comprised seven items, from which five items corresponded to CMAI Factor 2 “Physically non-aggressive behavior”. The items “hiding things” and “hoarding things” were not considered in the factor analysis reported in the CMAI manual.

Besides the fact that several previous studies did not include every item occurring in a percentage lower than 5% or 10% in the factor analysis [17,22,23,24], we considered that by disregarding some items the estimation of the clinical involvement would be biased. It is worth considering that, especially in longitudinal studies such as RECage, in which there were many follow-up visits over three years, the exclusion of some items would lead to missing useful clinical information. New symptoms may occur or earlier symptoms may improve or worsen over time; Therefore, we think that it is crucial to calculate all the items in the final scores. 

According to our results, there was no sharp distribution of the items on the retained factors, leading to some interpretation problems. Indeed, besides the fact that we excluded items with low frequency (≤10%) from the analysis, there was a great heterogeneity regarding the frequencies of some of the remaining items, with prevalence ranging from 14% to 78%. Specifically, the items “tearing things or destroying property”, “throwing things”, “hitting (including self)”, “grabbing onto people and pushing”, have a prevalence lower than 25%. 

We think that the recommendation to discard some items in the patients’ scoring of the CMAI with a prevalence of ≤5% raises a big problem. Indeed, if they are not reckoned in the three factors suggested in the CMAI analysis according to its manual, there is the risk of attributing the same scores to patients with and without some of the excluded items. Taking into account that the total of the 9 items not considered ranges from 9 to 63, it is possible to have a very different evaluation of the severity of the disease. In addition, this does not allow appropriate quantification of the pattern of the CMAI change over the time within the same person.

### 4.1. New Factors Proposal by Following a More Clinical Approach

We think that it would be useful to adopt a more sensible approach. A first suggestion could be to consider the excluded items of the 29 or 36 items of the CMAI in a separate cluster to be analyzed by itself. However, a more reasonable procedure would be to cluster all the CMAI items (versions of 29 or 37 items), according to a more “clinical approach”. Our proposal is that the 29 items of the CMAI be clustered as follows, according to De Vreese [34]:

Factor 1 according to CMAI manual was “Aggressive physical behavior”. In this section we could include the following eleven items “3—Spitting (F1)”, “7—Hitting (including self) (F1)”, “8—Kicking (F1)”, “9—Grabbing onto people (F1)”, “10—Pushing (F1)”, “11—Throwing things (NC)”, “14—Biting (F1)”, “15—Scratching (F1)”, “21—Hurt self or other (NC)”, “25—Tearing things or destroying property (F1)”, and “28—Making physical sexual advances (NC)”. All the above items/symptoms refer to behaviors that are quite aggressive and are expressed via a physical means. The items “throwing things”, “hurt self or other” and “making physical sexual advances”, which are not considered based on frequency criteria in factor one, should be included by following clinical criteria in this factor, since such behaviors have a totally aggressive and physically harmful impact to patient’s life or to other’s life. 

On the other hand, Factor 2, according to the CMAI manual comprise the “Physical non aggressive behavior” and we believe that it should include the following 10 items: “1—Pace, aimless wandering (F2)”, “2—Inappropriate dress or disrobing (F2)”, “16—Trying to get to a different place (F2)”, “17—Intentional falling (NC)”, “20—Eating/drinking inappropriate substances (NC)”, “22—Handling things inappropriately (F2)”, “23—Hiding things (NC)”, “24—Hoarding things (NC)”, “26—Performing repetitious mannerisms (F2)”, “29—General restlessness (F2)”. The items “intentional falling”, “eating/drinking inappropriate substances”, as well as “hiding and hoarding things”, that are not considered factors, are believed to be clinically categorized in physically non-aggressive behavior, since they are very annoying symptoms expressed via the body but are not harmful to anyone. 

Finally, Factor 3 which comprises the “Aggressive verbal behaviors” in our opinion could include the following 8 items: “4—Cursing or verbal aggression (F1)”, 5—Constant unwarranted request for attention or help (F3)”, “6—Repetitive sentences or questions (F3)”, “12—Strange noises (weird laughter or crying) (NC)”, “13—Screaming (F3)”, “18—Complaining (F3)”, “19—Negativism (F3)”, and “27—Making verbal sexual advances (NC)”. Regarding the items “strange noises” and “making verbal sexual advances”, besides the fact that they do not physically harm a person, are aggressive and annoying behaviors in a verbal way. 

It is worth mentioning that the three above reported factors comprise the items included in the CMAI Factors 1, 2, and 3, by including the items related to the pertinent behaviors not included in the CMAI factor analysis because of their low prevalence. Therefore, it is essential to suggest a CMAI scoring method that considers all twenty-nine items based on a sound clinical approach, as De Vreese first proposed [34].

Of course, the proposed “clinical structure” with three factors has to be validated in longitudinal studies in comparison to the “classical” three-factor CMAI structure reported in the CMAI manual in order to establish the more appropriate method for capturing the changes in symptomatology.

### 4.2. Study’s Strengths 

As mentioned above, the CMAI’s Factor Analysis was conducted as a part of the RECage study, which was a longitudinal and multicultural study that comprised 11 clinical centers from six European countries (Italy, Germany, France, Greece, Switzerland, and Norway) and included a sample of 505 patients. It should be stated that, generally speaking, the sample size for studies with Factor Analysis should be at least 300 participants, whilst the variables that are subjected to factor analysis should have at least 5 to 10 observations each [35,36]. Besides the fact that many studies have investigated the CMAI factor structure, several of them included a limited number of participants and samples from the same country. Several of them are also referred to in the CMAI manual [16,17,20,21,37,38]. 

Furthermore, a second strength concerns the suggestion of a new, more clinical approach to the CMAI assessment which includes all items, avoiding exclusion of them based on their reduced prevalence. This new approach is based on grouping the items regarding their clinical meaning; therefore, important information regarding the clinical course of dementia and the effectiveness of the treatment will not missed.

## 5. Conclusions

Our study identified a similar underlying construct of the CMAI reported in its pertinent manual with a three-factor model characterized by: (a) Aggressive behavior, (b) Physically non-aggressive behavior, and (c) Verbally agitated behavior. Of course, there are several studies that found and proposed a different factor structure, a fact that makes sense since the clinical population differs, especially when the samples come from different cultural environments. 

Furthermore, a confirmatory FA on our data did not confirm the three-factor structure proposed in the CMAI manual. We propose instead a three-dimensional model that includes all the CMAI items based on their clinical characteristics, aiming to include all items in the scoring procedure.

## Figures and Tables

**Table 1 brainsci-13-01025-t001:** Items of the CMAI questionnaire, in absolute and percentage frequencies of the PwD with BPSD of the RECage study.

	CMAI Items	Absolute Number	Percent
1	Pace, aimless wandering (F2)	271	53.66%
2	Inappropriate dress or disrobing (F2)	253	50.10 %
3	Spitting (F1)	37	** 7.33%
4	Cursing or verbal aggression (F1)	202	40.00%
5	Constant unwarranted request for attention or help (F3)	273	54.06%
6	Repetitive sentences or questions (F3)	398	78.81%
7	Hitting (including self) (F1)	84	16.63%
8	Kicking (F1)	45	** 8.91%
9	Grabbing onto people (F1)	103	20.40%
10	Pushing (F1)	111	21.98%
11	Throwing things (NC)	74	14.65%
12	Strange noises (weird laughter or crying) (NC)	168	33.27%
13	Screaming (F3)	172	34.06%
14	Biting (F1)	13	* 2.57%
15	Scratching (F1)	20	* 3.96%
16	Trying to get to a different place (F2)	203	40.20%
17	Intentional falling (NC)	23	* 4.55%
18	Complaining (F3)	320	63.37%
19	Negativism (F3)	341	67.52%
20	Eating/drinking inappropriate substances (NC)	49	** 9.70%
21	Hurt self or other (NC)	31	** 6.14%
22	Handling things inappropriately (F2)	169	33.47%
23	Hiding things (NC)	255	50.50%
24	Hoarding things (NC)	209	41.39%
25	Tearing things or destroying property (F1)	71	14.06%
26	Performing repetitious mannerisms (F2)	300	59.41%
27	Making verbal sexual advances (NC)	38	** 7.52%
28	Making physical sexual advances (NC)	30	* 5.94%
29	General restlessness (F2)	361	71.49%

Abbreviations: F1 = Factor 1—Aggressive behavior according to the CMAI manual; F2 = Factor 2—Physically non-aggressive behavior according to the CMAI manual; F3 = Factor 3—Verbally agitated behavior according to the CMAI manual; NC = Not considered in the Factor Analysis shown in the CMAI manual; * = Items with a prevalence less than 6% not considered in our first exploratory Factor Analysis on 25 items, reported in the CMAI Manual; ** = Items with a prevalence less than 10% not considered in our second exploratory Factor Analysis on 25 items.

**Table 2 brainsci-13-01025-t002:** Factor analysis on 25 items with a prevalence greater than 6%.

	CMAI Items	Physically Aggressive Behavior	Verbally Aggressive Behavior	Physically Non-Aggressive Behavior	Physically & Verbally Aggressive Behavior
1	Pace, aimless wandering (F2)	0.19719	0.15575	* 0.41093	−0.15400
2	Inappropriate dress or disrobing (F2)	0.12055	0.22691	* 0.48081	0.16414
3	Spitting (F1)	0.22244	0.01023	0.18102	* 0.27099
4	Cursing or verbal aggression (F1)	0.26895	0.18031	0.13978	* 0.55374
5	Constant unwarranted request for attention or help (F3)	0.13927	* 0.65967	0.12949	0.02939
6	Repetitive sentences or questions (F3)	−0.10439	* 0.38519	0.21625	0.21014
7	Hitting (including self) (F1)	* 0.77277	0.10124	0.02532	0.09326
8	Kicking (F1)	* 0.61399	0.07846	0.08388	0.02772
9	Grabbing onto people (F1)	* 0.61592	0.03500	0.15331	−0.00955
10	Pushing (F1)	* 0.71998	0.03020	0.05529	0.21056
11	Throwing things (NC)	* 0.53871	0.10127	0.05074	0.30253
12	Strange noises (weird laughter or crying) (NC)	−0.03523	* 0.41407	0.21909	0.27350
13	Screaming (F3)	0.31439	0.24346	0.13015	* 0.46173
16	Trying to get to a different place (F2)	0.26892	0.18218	* 0.55827	−0.13590
18	Complaining (F3)	0.02801	* 0.75179	0.07885	0.18285
19	Negativism (F3)	0.02365	* 0.68209	0.12769	0.08561
20	Eating/drinking inappropriate substances (NC)	0.04846	0.04709	* 0.40158	0.07022
21	Hurt self or other (NC)	* 0.39506	−0.07391	0.15834	0.02246
22	Handling things inappropriately (F2)	0.06293	0.10589	* 0.54607	0.12447
23	Hiding things (NC)	0.06381	0.19066	* 0.54449	0.18026
24	Hoarding things (NC)	0.01066	0.09272	* 0.49180	0.22748
25	Tearing things or destroying property (F1)	0.26794	−0.00636	* 0.29928	0.06883
26	Performing repetitious mannerisms (F2)	0.06730	0.40019	* 0.47852	−0.03655
27	Making verbal sexual advances (NC)	0.02267	0.10211	0.02755	* 0.37694
29	General restlessness (F2)	0.19876	* 0.56196	0.38329	0.11321

Abbreviations: F1 = Factor 1—Aggressive behavior according to the CMAI manual; F2 = Factor 2—Physically non-aggressive behavior according to the CMAI manual; F3 = Factor 3—Verbally agitated behavior according to the CMAI manual; NC = Not considered in the Factor Analysis according to the CMAI manual; * identify the strongest factor loading for each item.

**Table 3 brainsci-13-01025-t003:** Factor analysis on 20 items with a prevalence greater than 10.0% and constrained to three factors.

	CMAI Items	Verbally Aggressive Behavior	Physically Aggressive Behavior	Physically Non-Aggressive Behavior
1	Pace, aimless wandering (F2)	0.06164	0.13034	* 0.45015
2	Inappropriate dress or disrobing (F2)	0.23655	0.16734	* 0.48130
4	Cursing or verbal aggression (F1)	0.28903	* 0.39850	0.13678
5	Constant unwarranted request for attention or help (F3)	* 0.59311	0.13540	0.17165
6	Repetitive sentences or questions (F3)	* 0.42776	−0.02826	0.22319
7	Hitting (including self) (F1)	0.04230	* 0.69903	0.08623
9	Grabbing onto people (F1)	−0.02309	* 0.57355	0.16930
10	Pushing (F1)	−0.00914	* 0.79117	0.09555
11	Throwing things (NC)	0.13011	* 0.59895	0.06872
12	Strange noises (weird laughter or crying) (NC)	* 0.47430	0.05813	0.20661
13	Screaming (F3)	0.32045	* 0.44645	0.15183
16	Trying to get to a different place (F2)	0.09658	0.20239	* 0.57706
18	Complaining (F3)	* 0.78628	0.09983	0.07686
19	Negativism (F3)	* 0.67929	0.04533	0.13595
22	Handling things inappropriately (F2)	0.14164	0.10178	* 0.49019
23	Hiding things (NC)	0.21339	0.08908	* 0.56581
24	Hoarding things (NC)	0.15485	0.03128	* 0.49174
25	Tearing things or destroying property (F1)	0.01831	* 0.26152	0.25112
26	Performing repetitious mannerisms (F2)	0.34529	0.02836	* 0.51796
29	General restlessness (F2)	* 0.52848	0.22632	0.42558

Abbreviations: F1 = Factor 1—Aggressive behavior according to the CMAI manual; F2 = Factor 2—Physically non-aggressive behavior according to the CMAI manual; F3 = Factor 3—Verbally agitated behavior according to the CMAI manual; NC = Not considered in the Factor Analysis according to the CMAI manual * identify the strongest factor loading for each item.

## Data Availability

Data are available from Bruno Cesana and Carlo Alberto Defanti.

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
