# Peer review of "A Confirmatory and an Exploratory Factor Analysis of the Cohen-Mansfield Agitation Inventory (CMAI) in a European Case Series of Patients with Dementia: Results from the RECage Study"

_brainsci, 2023, doi:10.3390/brainsci13071025_

Round 1

Reviewer 1 Report

In this paper, authors examined the application of CMAI in a sample of 505 patients from the RECage study. In details, they performed a factor analysis to identify potential underlying factors related to CMAI items (exploratory factor analysis) as well as testing the factors proposed by CMAI study (confirmatory factor analysis). There are no major technical/analysis-related issues within the paper.

Some minor suggestions should be taken in consideration:

In methods, please add which extraction method you used for Factor analysis (e.g principal component analysis (PCA))

A spellcheck should be perfomed on the manuscript

e.g. section 2.2 partecipants, seems to have some minor issues. Also, in section 2.2 please at the end add the reference to Poptsi et al. paper.

In section 3. results: it's not clear in the text if all the 9 items were excluded from the analyses. According to table 1 data, the 4 items with prevalence <5% were excluded from EFA1 while all  9  were excluded from EFA2. is this interpretation correct? Also, item 28 in table 1 has prevalence 5.94% but it is labeled as item with less or about 5%. It may be better to label the items as items with prevalence <6%.

section 3.1 and 4. some parts of the text are not clear.

I have no more suggestion/observation. I thank the authors for their work.

just some minor errors within the manuscript (e.g. lane 411 having exceeding should be having exceed). Nothing too serious but a spellcheck of the manuscript is recommended 

Author Response

Dear reviewer,

Below you may find our responses regarding your comments, after the first review. The format below adheres to the format:

(R)eviewer(Reviewer number)(C)omment(Comment number)

(R)eviewer(Reviewer number)/(R)esponse

#1 Reviewer’s Comments

R1C1: In methods, please add which extraction method you used for Factor analysis (e.g principal component analysis (PCA)

R1R1: We thank the reviewer for giving us the opportunity of reporting that the Principal Component Analysis (PCA) has been the method used in our factor analysis. The part of the statistical methods (section 2.5) has been changed as: Several rotation methods (varimax, promax, etc.) have been used in EFA, after the factor extraction with the Principal Component Analysis (PCA), to have better differentiation of the factor loadings [31-33].

R1C2: A spellcheck should be performed on the manuscript. e.g. section 2.2 participants, seem to have some minor issues. Also, in section 2.2 please at the end add the reference to Poptsi et al. paper.

R1R2: We thank the reviewer for the comment, we have checked the language, having corrected the minor issues. We have also added the reference of Poptsi et al.

R1C3: In section 3. results: it's not clear in the text if all the 9 items were excluded from the analyses. According to table 1 data, the 4 items with prevalence <5% were excluded from EFA1 while all 9 were excluded from EFA2. Is this interpretation correct? Also, item 28 in table 1 has prevalence 5.94% but it is labeled as item with less or about 5%. It may be better to label the items as items with prevalence <6%.

R1R3: In this paper we have performed 2 different exploratory factor analyses (EFA). Indeed, the first EFA was performed on the 25 items of the total 29 items of CMAI, since we discarded from the analysis the 4 items that were not considered in the three-factors structure of the CMAI reported in the CMAI manual. Furthermore, we have also carried out an extra Factor Analysis on 20 items, by excluding the items with a prevalence less than 10%. As regards the item no 28 that was not included in the factor analysis in the legend of the table has been written: “Items with a prevalence less or about 5% not considered in…” in any case it is possible to change as <6% as the reviewer suggested. The fact is that the item 28 was also excluded from the factor analysis of the CMAI shown in the manual.

R1C4: section 3.1 and 4. some parts of the text are not clear.

R1R4: Thank the reviewer for the comment. Section 3.1 was rewritten as:

After 12 iterations the convergence has been reached. The model fit 2 was 960.156 (df = 167, p ≤0.0001) statistically rejecting the confirmatory factor model of the CMAI. Indeed, the Root Mean Square Error of Approximation (RMSEA) was 0.2289 greater than the conventional 0.05 value for a good model fit. The Standardized Root Mean Square Residual (SRMSR) was 0.0867, not closed to the conventional 0.05 value for a good model fit. In addition, Bentler’s comparative fit index was 0.7262 much lower than the required value of at least 0.90 leading to conclude for a very poor model fit. So, taking into account that all the above four criteria consistently testify to an inadequate model fit, it is possible to conclude that the CMAI three- factors model proposed in the CMAI manual was not adequate to our data.

Also the discussion was extensively reviewed and rewritten

I have no more suggestion/observation. I thank the authors for their work.

R1C5: Just some minor errors within the manuscript (e.g. lane 411 having exceeding should be having exceed). Nothing too serious but a spellcheck of the manuscript is recommended 

R1R5: Thank you, we have performed the spellcheck.

Reviewer 2 Report

The study examined factor analysis using both exploratory and confirmatory factor analysis methods on the Cohen-Mansfield Agitation Inventory. Overall, the study is straightforward and well-written. However, I have only a few comments.

  1. In section 2.5 (Statistical Methods), the author needs to provide more detailed information about the statistical analysis methods used in the study. For example, in the case of confirmatory factor analysis (CFA), the hypothesis (3-factor model) should be clearly stated. It is also unclear how many items were included in the CFA (section 3.1). Additionally, the author should briefly explain the specific technique used for the CFA model fit statistics. Similar clarifications are needed for the exploratory factor analysis (EFA) section.

  2. I noticed that in section 3.1, it is possible that CFA was performed using all 29 items of the CMAI. However, later in the EFA section, the author used 20 and 25 items for analysis. It would be helpful to understand why the author did not analyze the CFA by excluding items with low prevalence as well.

  3. Lastly, I suggest that the author should add more information about the benefits of this study for clinical practice. It would be interesting to know whether the use of a 3 or 4 factor model has any implications for clinical practice. Additionally, it would be beneficial to include specific suggestions for further studies regarding the factor validity of the Cohen-Mansfield Agitation Inventory (CMAI).

  4. Typo: Page 5, Line 243: "Furthermore, the mean NPI score the mean score was..."

Author Response

Dear reviewer,

Below you may find our responses regarding your comments, after the first review. The format below adheres to the format:

(R)eviewer(Reviewer number)(C)omment(Comment number)

(R)eviewer(Reviewer number)/(R)esponse

#2 Reviewer’s Comments

R2C1: In section 2.5 (Statistical Methods), the author needs to provide more detailed information about the statistical analysis methods used in the study. For example, in the case of confirmatory factor analysis (CFA), the hypothesis (3-factor model) should be clearly stated.

It is also unclear how many items were included in the CFA (section 3.1). Additionally, the author should briefly explain the specific technique used for the CFA model fit statistics. Similar clarifications are needed for the exploratory factor analysis (EFA) section.

R2R1: We thank the reviewer for giving us the opportunity of clarify our analysis. The hypothesis of the CFA has been added in section 2.5 together with the considered number of items (20).

 R2C2: I noticed that in section 3.1, it is possible that CFA was performed using all 29 items of the CMAI. However, later in the EFA section, the author used 20 and 25 items for analysis. It would be helpful to understand why the author did not analyze the CFA by excluding items with low prevalence as well.

R2R2: Many thanks for your interesting question. This has been answered in the previous question. Here we would like to say that our main interest in the CFA was to test the hypothesis about the CMAI factor structure on 20 items reported in the CMAI manual. Furthermore, we performed two EFA on 25 items by eliminating the items with a prevalence of <6% and on 20 items by eliminating the items with a prevalence of <10%.

R2C3: Lastly, I suggest that the author should add more information about the benefits of this study for clinical practice. It would be interesting to know whether the use of a 3 or 4 factor model has any implications for clinical practice. Additionally, it would be beneficial to include specific suggestions for further studies regarding the factor validity of the Cohen-Mansfield Agitation Inventory (CMAI).

R2R3: Many thanks for your interesting suggestion. We added a sentence in the Discussion about the 3 or 4 factor structure: However, the fact that these three items are already considered in the CMAI Factor 1 “aggressive behavior” limits the relevance of our finding allowing us to conclude that a further subdivision in 4 factors cannot have any clinical relevance”.

Furthermore, we added a sentence at the end of the section “4.1. New factors proposal by following a more clinical approach” about the proposal for new studies:

“Of course, the proposed “clinical structure” with three factors has to be validated in longitudinal studies in comparison to the “classical” three factors CMAI structure reported in the CMAI manual in order to establish the method more appropriate to capture the changes in the symptomatology”.

R2C4: Typo: Page 5, Line 243: "Furthermore, the mean NPI score the mean score was..."

R2R4: The typographic error has been corrected.